Epidemiology and transmission of hepatitis A in Shaanxi (western China) after more than ten years of universal vaccination

Hu Xiaotong huxiaotongla@163.com 1
Hu Weijun 2
Dong Yuanyuan 2
Lu Xuan 3
Xu Fujie 4
Zhang Shaobai maolyzhang@163.com 2
1 The Affiliated Hospital of Guizhou Medical University , Guizhou , China
2 Shaanxi Provincial Center for Disease Control and Prevention , Xi’an , China
3 Department of Geriatric and Emergency Medicine, The First Affiliated Hospital, School of Medicine, Zhejiang University , Hangzhou , Zhejiang , China
4 Bill& Melinda Gates Foundation , Beijing , China
Tsuzuki Shinya
Electronic publication date: 2023 Nov 13
Publication date: 2023
Volume: 11
Electronic Location ID: e16305
Received 2023 Jul 10; Accepted 2023 Sep 26
Copyright: ©2023 Hu et al.
Copyright year: 2023
Copyright holder: Hu et al.
License: This is an open access article distributed under the terms of the Creative Commons Attribution License, which permits unrestricted use, distribution, reproduction and adaptation in any medium and for any purpose provided that it is properly attributed. For attribution, the original author(s), title, publication source (PeerJ) and either DOI or URL of the article must be cited.
License URL: https://creativecommons.org/licenses/by/4.0/

Keywords: Hepatitis A, HepA, HAV, Seroprevalence, China, Endemicity, Transmission risk

Funding: National Science and Technology Major Project of China No. 2018ZX10201002 Doctor Start-up Fund of Affiliated Hospital of Guizhou Medical University NO.GYFYBSKY-2023-24 This work was supported by the National Science and Technology Major Project of China (No. 2018ZX10201002) and the Doctor Start-up Fund of Affiliated Hospital of Guizhou Medical University (NO.GYFYBSKY-2023-24). The funders had no role in study design, data collection and analysis, decision to publish, or preparation of the manuscript.

==============================
Background

Hepatitis A (HepA) vaccination and economic factors can change the epidemiology of HepA. In China, the implementation of free vaccination for children under 1.5 years of age in 2008 has resulted in a decline in the overall incidence of HepA. Nevertheless, further investigation is required to comprehensively understand the epidemiological patterns of HepA in economically disadvantaged regions of China.

Method

In this study, we evaluated the incidence, seroprevalence, and transmission characteristics of HepA in Shaanxi with less economically developed. We obtained data on reported cases of HepA from 2005 to 2020. Blood samples from 1,559 individuals aged 0 to 60 years were tested for anti-hepatitis A (HAV) antibodies. A questionnaire survey and blood sample collection were conducted in two sentinel sites from 2019 to 2021.

Result

Between 2008 to 2020, the number of reported cases of HepA decreased from 3.44/100,000 person-years to 0.65/100,000 person-years, indicating an 81.1% decrease, which was particularly pronounced among younger age groups (0–19 years). From 2015–2020, infections were more likely to occur in people in their 40s and those over the age of 60. Farmers were still the most common occupation of HepA in the last decade. The results of the serological investigation showed the highest anti-HAV seroprevalence was observed in adults aged 39–60 years (94.6%) and those aged 28–38 years (87.8%). The 10–15 years group had the lowest seroprevalence at 49.3%. During the study period, a total of 22 cases were reported by sentinel sites, but the common risk factors (like raw food exposure, travel history, and closed contact with patients) were not identified.

Conclusion

Given the greater severity of illness in the adult population and the ambiguous transmission routine, enhanced surveillance for HepA and evaluations that identify feasible approaches to mitigate the risk of HAV transmission are urgent priorities.

Introduction

As a foodborne disease, the incidence of hepatitis A (HepA) is highly associated with environmental sanitation and economic status (Aggarwal & Goel, 2015). It is usually caused by drinking unclean water or by eating rural food that was contaminated by the hepatitis A virus (HAV), and it appears to be more severe in older populations and populations with underlying medical conditions (WHO, 2010). The prevalence of HepA in China has experienced a significant decline after the introduction of the HepA vaccine into the Expanded Program on Immunization as a complimentary childhood vaccine (Ren et al., 2017), The recommended immunization schedule entails either one dose of live attenuated HepA at 18 months or two doses of inactivated HepA, with the initial dose administered at 18 months and a subsequent dose at 24 months (Wang et al., 2020). HepA was initially prioritized for the urban region and was expanded to province-wide use in Shaanxi by 2010. The declining morbidity of HepA among children, leads to increased susceptibility in older generations who are more likely to suffer from the more severe disease (Wang et al., 2020), highlighting the significance of HepA as an important disease in terms of public health and its burden on the medical system.

Previous seroprevalence studies clearly indicate that a sizable adult population lacks immunity to HAV (Wang et al., 2020), but the percentage and age patterns varied greatly from one province to another across China, higher immunity levels were found in Xinjiang and lower in Shanghai (Zhang et al., 2017; Zhu et al., 2013). Compared to economically developed provinces, the availability of recent epidemiological data for less developed provinces is limited or severely limited. Shaanxi, one of the biggest provinces in western China, had a population of 40 million in 2020. As a less advanced province, its transmission characteristics of HepA are not only affected by migration and food distribution within China but also affected by the imported food from high-endemicity countries and citizens who travel to countries with a high incidence of HepA. As reported in the epidemiological data of Shaanxi province, we evaluated the prevalence of hepatitis A cases. Vaccination programs were also evaluated with serum anti-HAV IgG to determine their impact on hepatitis A incidence.

Materials & Methods

Data source and analysis

All cases of HepA reported during 1990-2020 from China’s National Notifiable Disease Report System (NNDRS) were collected, confirmed cases were determined based on diagnostic criteria issued by the Chinese Ministry of Health in 2008 (Ren et al., 2017). The demographic information of individual cases reported between 1990 and 2004 was not available. Immunization information of HepA originated from the Shaanxi provincial planned immunization information system, which was established in 2010. An analysis of the data was conducted using SPSS software version 16.0 (SPSS Inc., Chicago, IL, USA). Data were presented as means and percentages for continuous variables and categorical variables, respectively. The χ2 test, Fisher’s precise test, or Wilcoxon rank-sum test were then used, depending on the situation. The multiple comparisons were corrected using the Bonferroni correction for chi-square tests. Statistical significance was considered at a level of 0.05 (P <  0.05).

Seroprevalence study

As a retrospective study, all serum samples were retrieved from a provincial serological study in Shaanxi. This provincial serological study was conducted in 2017 by Shaanxi Provincial Center for Disease Control and Prevention and this provincial was designed to assess the prevalence of Hepatitis B. We use the residual blood from that serological survey and analyzed samples for anti-HAV antibody levels. The provincial study employed a 2-stage cluster random sampling technique to select participants from the 10 counties. Subsequently, 30 villages were chosen in the 10 counties (two or three per county) using probability proportional to size sampling (PPS). Individuals were then divided into five groups by birth year: (1) 2009–2016: years after HepA vaccines were integrated into the expanded program on immunization (EPI). (2) 2003-2008: L-HepA and I-HepA vaccines were both offered at an out-of-pocket expense. (3) 1992–2002: L-HepA vaccine was available. (4) 1980–1991: years before the HepA vaccine was issued. (5) 1958–1979: years before rapid urbanization and improved sanitation in Chinese populations. After individuals were divided into different age groups, they were randomly selected according to the designed sample size. The estimated sample size was calculated for each age group (Table S1). Anti-HAV antibodies were detected by commercial kits (ARCHITECT HAVAb-IgG; Abbott, Wiesbaden, Germany) according to the manufacturer’s instructions. Samples with the signal-to-cut-off (S/CO) ratio ≥ 1.00 were considered positive for anti-HAV IgG.

Study population in sentinel sites

We launched a prospective study in sentinel sites, patients with HepA presenting at the NNDRS from 2019 to 2021 were screened by trained personnel. HepA cases must be symptomatic with a discrete onset of symptoms consistent with acute viral hepatitis, and jaundice or elevated serum aminotransferase levels, and anti-HAV IgM must test positive. Serum samples and epidemiological data from reported cases were collected, and specific procedures were reported in our previous study (Hu et al., 2021).

Ethical aspect

In order to better demonstrate the transmission characteristics of HepA, we collected data from three aspects: historical incidence data of HepA in Shaanxi, the seroprevalence of anti-HepA antibodies in the general population, and case reports from sentinel hospitals. The first part is the incidence of HepA, it mandates that medical and health institutions at all levels must report HepA to the NNDRS based on the Law of the People’s Republic of China on the Prevention and Treatment of Infectious Diseases. This system does not collect data on the epidemiological details of the cases and the molecular information of pathogens. The second part is the seroprevalence of anti-HepA antibody in the general population. In accordance with the research ethics committee of the first affiliated hospital, College of Medicine, Zhejiang University (Number (2019,998), Number (2019,1482)), and considering the minimal risks involved as well as the unavailability of contact information for the original data subjects (telephone numbers or addresses were not collected in the provincial surveys), the requirement for informed consent was waived for the seroprevalence study. The third component of this study involved a prospective investigation conducted at sentinel sites, wherein serum samples for sequencing purposes and epidemiological data were gathered from patients diagnosed with HepA at designated hospitals between the years 2019 and 2021. In compliance with the principles outlined in the Declaration of Helsinki, written or oral consent was obtained from the patients who were included in the reported cases at the sentinel sites.

Results

Epidemiological characteristics of reported cases in Shaanxi

As shown in Fig. 1, the annual incidence of HepA was 55.31/100,000 person-years before the HepA vaccine was introduced, then it declined significantly. Since 2008, HepA vaccines have been included in the planned immunization program, and the number of reported cases of HepA gradually decreased, from 3.44/100,000 person-years in 2008 to 0.65/100,000 person-years in 2020. From 2008 to 2019, a total of 4.7 million children were vaccinated, until 2019, almost all children aged 1.5–9 years in Shaanxi were vaccinated for HepA (Fig. 1). In the years before the HepA vaccine was included as a free vaccine in the EPI system (2005–2008), the HepA incidence among younger age groups (0–19 years) accounted for the largest proportion (31.9%–42.9%), In 2013, the number of reported cases aged 0–19 years declined sharply to 10.4%, and in 2020, it had declined to less than 2% (Fig. 2A). This phenomenon was more pronounced in children aged 0–9 years (Fig. 2B). This was accompanied by individuals aged 40–49 years who occupied a relatively larger proportion of the total. From 2015–2020, infections were more likely to occur in people in their 40s and those over the age of 60. Infections were initially mainly diagnosed in males (63%), but after 2018 there was almost a 50/50 split between males and females (Fig. 3). In the pre-vaccine era, hepatitis A incidence was highest among students, children in kindergarten, and farmers. However, the proportion of students and children in kindergarten decreased significantly after 2012, which was nearly 80% lower than that of the peak. In contrast, farmers remained the predominant occupation associated with HepA. It is noteworthy that there has been a gradual increase in the incidence of HepA among service providers over time (Fig. 4).

Figure 1 The incidence rate and vaccination coverage of hepatitis A in Shaanxi, China, 1990–2020.

L-HepA, live attenuated HepA vaccine; I-HepA, inactivated HepA vaccine; EPI, Expanded Program on Immunization.

Figure 2 Age distribution of patients with hepatitis A in Shaanxi, China, 1990–2020.

(A) Age distribution of patients with hepatitis A in Shaanxi, China, 1990–2020. (B) The proportion of cases aged under 10 years old (children%) by year.

Figure 3 Gender distribution of Hepatitis A cases reported in Shaanxi, China, 1990–2020.

Figure 4 Occupational distribution of Hepatitis A cases reported in Shaanxi, China, 1990–2020.

Seroprevalence study in Shaanxi

A total of 1,559 individuals were enrolled in this study, of those, 589 individuals were from urban areas, and were 970 from rural areas. The male/female ratio was 0.85:1. The highest anti-HAV seroprevalence was observed in adults aged 39–60 years (94.6%) and those aged 28–38 years (87.8%). The 10–15 years group had the lowest seroprevalence at 49.3%. No statistically significant differences were found in the seroprevalence of males and females (P = 0.325). But the distribution of seroprevalence was remarkably different for individuals from urban and rural areas (P < 0.000) (Table 1).

Table 1 The seroprevalence of anti-HAV IgG antibodies in Shaanxi in 2017.

	Sample size (N)	Seroprevalence (%)	Pearson’s x2	P	
Age groups:					
2∼9	345	80.9			
10∼15	414	49.3*		<0.05	
16∼27	402	59.0*		<0.05	
28∼38	286	87.8			
39∼60	112	94.6*		<0.05	
Urban/rural:					
Urban	589	74.7			
Rural	970	65.7	14.000	<0.000	
Gender					
Male	715	67.8			
Female	844	70.1	0.967	0.325	
Total	1,559	69.1			
Notes.

The chi-square test was done with comparing to 2–9 years group.

Characteristics of hepatitis A cases in Shaanxi

During the study period, a total of 22 cases were reported by sentinel sites (Table 2). Of those, 11 cases (50%) were males, with a median age of 61.5 years old. The majority of cases were observed in urban residents. The primary presenting symptoms were fever (18.2%) and abdominal pain (18.2%), followed by loss of appetite (13.6%), jaundice (9.1%), and nausea (4.5%). The median value of ATL and TBil are 148 U/L and 20.2 umol/L, respectively. Two asymptomatic patients were reported by annual health examination. The HepA incidence in urban areas was higher than that in rural areas. Approximately 90% of reported cases did not live alone, yet no instances of household transmission were identified. In terms of possible transmission routes, the proportion of eligible samples for sequencing analysis was zero. According to the epidemiological questionnaire, common risk factors (like raw food exposure, travel history, and closed contact with patients) were not reported among those patients.

Table 2 Patient characteristics in Shaanxi, 2019-2021.

Characteristics		
Median age (range), years	61.5 (35–89)	
Sex (%)
Female
Male	
11 (50.0%)
11 (50.0%)	
Residence (%)
Rural
Urban	
7 (31.8%)
15 (68.2%)	
Initial symptoms (%)
Fever
Nausea
Jaundice
Abdominal pain
Loss of appetite
Others	
18.2%
4.5%
9.1%
18.2%
13.6%
36.4%	
ALT[(median(range)], U/L	148 (11.4–1852)	
TBIL[(median(range)], umol/L	20.2 (1.7–325.1)	
No. household contacts[(median(range)]	3 (0–5)	
No. asymptomatic cases.	2	
Possible transmission route (%)
Raw seafood exposure
Recent travel history
Household contact	

0 (0%)
0 (0%)
0 (0%)	

Discussion

It is well established that socioeconomic improvements are highly related to the incidence of HepA and the rate of HepA vaccination (Aggarwal & Goel, 2015; Zhang et al., 2008). In recent decades, GDP per capita increased rapidly in China, but with significant differences in the rate of increase. Provinces were categorized into three regions—eastern, central, and western indicating high (eastern) to low (western) socio-economic development. Similarly, the morbidity of HAV infection decreased in all three regions with the highest incidence shifting from the eastern region to the western region (Sun et al., 2019). However, the passive reporting system of HepA ignored asymptomatic infections, incidence rates could be underestimated by only a single aspect. For this reason, seroprevalence studies can provide a more accurate picture of the circulation of HAV. Previous research estimated that China has decreased from a high to an intermediate HAV endemicity (Aggarwal & Goel, 2015; Wang et al., 2020). This study provides a new aspect of HAV transmission characteristics in China under a similar vaccination policy among provinces with different economic statuses.

First, we collected details of reported cases in Shaanxi province with a less developed economy and a low rate of HepA vaccination among adults. Then we also performed a serological study since seroprevalence in the adult population is very much influenced by the time during which the overall level of endemicity in a certain region shifted from high to low. We found the prevalence of HepA decreased over time, especially for children and adolescents. It is credited mostly to HepA vaccination, but also to public awareness, improved social hygiene, and improvements in sewage treatment and water quality. However, the overall decrease in incidence was accompanied by a progressive rise in the average age of reported cases. This may be a result of the inclusion of the EPI only targeting children older than 18 months. This phenomenon has been observed in many countries where EPI for children was deployed (Jacobsen, 2018). While farmers and individuals who live in rural areas are still the key populations of HepA, the probable reasons were: using well water as a source of drinking, and a relatively compromised sewage treatment system (Chen et al., 2011). Although the total incidence of HepA dramatically decreased (Fig. 1), the estimated anti-HAV seroprevalence rate among individuals aged 30 was still high, indicating the continuous local circulation of HAV. Unlike other studies from either developing (Aggarwal & Goel, 2015) or developed regions (Carrillo-Santisteve et al., 2017), Shaanxi presented a U-shaped pattern of age-specific anti-HAV seroprevalence. Individuals born prior to the national introduction of the HepA vaccine (1992–2008) had lower anti-HAV seroprevalence than earlier (>87.8%) and later birth cohorts (80.9%). There are several contributing factors: first, as a developing country, China used to be a high HAV endemicity country (Sun et al., 2018), the anti-HAV seroprevalence usually increases with age, and most senior citizens were supposed to have acquired immunity through natural infection at an early age. In Shaanxi, more than 90% of individuals aged 28 years and over has immunity to HAV. Similar findings were observed across nearly all provinces in China (Wang et al., 2020). Second, the years from 2008 to 2011 form the transitional period for HepA integration into the EPI in China. Our study revealed that in Shaanxi, the target population achieved a coverage rate exceeding 99%, while individuals born after 2008 exhibited an anti-HAV seroprevalence rate of 80.9%. This robust evidence underscores the significant impact of HepA vaccination on the distribution of anti-HAV seroprevalence. Third, vaccinated children would not be efficient transmitters and represent a reservoir of HAV in the community, along with improvement of the social-economic situation and public education, fewer individuals would expose to HAV before they reach adolescence or adulthood. This may explain why unvaccinated individuals born between 1992 to 2008 had the lowest anti-HAV seroprevalence rate.

In terms of transmission routine, we had a low yield rate of HAV RNA from reported cases, a possible reason we have explained in our previous study, one potential explanation for the observed low rate could be attributed to the heightened sensitivity but limited specificity of the case reporting criteria in accurately identifying genuine HepA cases (Hu et al., 2021). Therefore, we are unable to find out the genomic relationship between those cases. The possible causes for this phenomenon are as follows: first, due to the limitation of our sample size, we cannot identify the actual transmission routine of HepA. In outbreak settings, without the inclusion of testing of all contacts, the identification of chains of transmission may be incomplete. Secondly, recall bias may have occurred because we asked interviewees to recall eating habits. some people failed to recognize that semi-cooked seafood still can contain pathogens, and cooking food fully and preparing raw and cooked food separately were always the key messages to avoid food-borne diseases. There was evidence of community spread in the study, and the source of the infection was unknown to the majority of participants. In order to have effective control, contact tracing and stopping transmission from contacts are essential.

Achieving a low incidence of HepA will require not only maintaining high coverage levels in the childhood vaccination program but implementing tailored strategies for provinces/municipalities with different endemicity. Generally, immunization program implementation with a catch-up strategy may be needed to avoid a possible future HAV outbreak. In this study, we demonstrated a U-shaped curve of age-specific HAV susceptibility prevalence in Shaanxi, where those born between 1980–2008 were most susceptible. However, a catch-up vaccination among young adults may not be cost-effective in high-endemicity provinces/municipalities where susceptibility in adults was low. In the case of Xinjiang, as well as the rural areas of many provinces/municipalities, more efforts need to be done for strengthening routine childhood immunization programs since timely vaccination for children was low (Liu et al., 2020; Sun et al., 2019). In addition, studies have demonstrated that anti-HAV seroprevalence was higher in rural areas than in urban areas, possible reasons could be that people are still drinking underground water or spring water in some rural areas (Chen et al., 2011; Ning, Zhengyi & Chaozhong, 2016), and use unsafe sanitation facilities which can facilitate HAV from the waste find their way into the local water supply (Kaifa et al., 2018; Kun, Hongying & Lunhong, 2018). Therefore, it is also important to promote better sanitation and safe drinking water. In provinces/municipalities with moderate endemicity, it is imperative to closely monitor the transmission characteristics of HAV in order to assess the effectiveness of preventive measures aimed at limiting risky behaviors. For instance, laboratory-based surveillance for HepA should be developed, allowing for early detection and precision investigation of outbreaks, describing transmission patterns, and better-targeted disease control (Hu, Collier & Xu, 2019). Evaluations that identify feasible approaches to mitigate the risk of HAV outbreaks due to the immunity gap are urgent priorities.

Conclusion

The considerable reductions in HepA incidence in Shaanxi, especially among children, represent the remarkable success of the introduction of a planned immunization program for HepA. But, given the heterogeneity of HAV endemicity levels and greater severity of illness in the adult population, enhanced surveillance for HepA, pilot testing catch-up campaigns, and cost-effectiveness evaluation study of those strategies to mitigate outbreaks are imperative.

Supplemental Information

Supplemental Information 1 Raw data for seroprevalence of hepatitis A in a healthy population from Shaanxi

Click here for additional data file.

Supplemental Information 2 The epidemiological characteristics of reported hepatitis A cases from China’s National Notifiable Disease Report System in Shaanxi

Click here for additional data file.

Supplemental Information 3 Historical data was retrieved from published articles (Kong et al., 2017, Yan et al., 2019)

# sample size was calculated by following equator (N = P*(1-P)*z2/MOE2), then this number was multiplied by 1.2 , in which z was type I error (α = 0.05), P was prevalence, and MOE was precision (confidence interval/2). Prevalence of Hepatitis A was estimated based on previous seroprevalence study, considering the efficiency of cluster sampling, the sample size was multiplied by 1.2 in the final version.

Click here for additional data file.

We would like to thank all the relevant health workers who took part in the collection samples from CDC in this study.

Additional Information and Declarations

Competing Interests

Author Contributions

Human Ethics

Data Availability

Fujie Xu is employed by Bill & Melinda Gates foundation after the study was finished.

Xiaotong Hu conceived and designed the experiments, analyzed the data, prepared figures and/or tables, authored or reviewed drafts of the article, and approved the final draft.

Weijun Hu performed the experiments, analyzed the data, prepared figures and/or tables, authored or reviewed drafts of the article, and approved the final draft.

Yuanyuan Dong performed the experiments, prepared figures and/or tables, authored or reviewed drafts of the article, and approved the final draft.

Xuan Lu performed the experiments, prepared figures and/or tables, and approved the final draft.

Fujie Xu conceived and designed the experiments, authored or reviewed drafts of the article, and approved the final draft.

Shaobai Zhang conceived and designed the experiments, analyzed the data, authored or reviewed drafts of the article, and approved the final draft.

The following information was supplied relating to ethical approvals (i.e., approving body and any reference numbers):

The research ethics committee of the first affiliated hospital, College of Medicine, Zhejiang University approved the study protocol (Number(2019,998), Number (2019)1482).

The following information was supplied regarding data availability:

The raw measurements are available in the Supplementary Files.

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
