# Peer review of "Epidemiology and transmission of hepatitis A in Shaanxi (western China) after more than ten years of universal vaccination"

_PeerJ, doi:10.7717/peerj.16305_

## Round 0.1 · original submission · Major Revisions

In addition to the concerns raised by the reviewers, the manuscript does not include explanations about their statistical analysis. The authors should describe how they analyzed the data in "Methods" section.

Reviewer 1 ·

Basic reporting

The manuscript is well-structured and clear to follow. Relevant information about the study setting (Shaanxi province) could be moved or included in a "setting" section in the Methods section.

The references are good and are sufficient for supporting the study. One more reference should be added, if available, as mentioned in the review below.

Experimental design

The investigators report a study of hepatitis A epidemiology and age-specific seroprevalence in Shaanxi province using four data sources: the Nationally Notifiable Disease Reporting Surveillance System (1990-2020), the Immunization Information System in Shaanxi, testing blood from a serological survey (n=1,559) for anti-hepatitis-A antibody, and evaluating hepatitis A cases in sentinel hospitals. They relate their analyses to HAV vaccine availability and introduction of HAV vaccine into the EPI system as a routine vaccine. They found that the incidence of hepatitis A decreased by 81% between 2008, the year at which HepA vaccine was introduced to EPI, and 2020. That the age of infection increased after vaccine availability and introduction, that most cases were among farmers, and that seroprevalence was highest among people in their 40s and 50s and was lowest among children 10-15 years of age. They concluded that “The considerable reductions in HepA incidence in Shaanxi, especially among children, represent a remarkable success of the introduction of a planned immunization program for HepA. But, given the heterogeneity of HAV endemicity levels and greater severity of illness in the adult population, enhanced surveillance for HAV infection and evaluations that identify feasible approaches to mitigate the risk of HAV transmission are urgent priorities.”

Validity of the findings

Evaluating the impact of vaccines and socioeconomic development on the epidemiology of HAV infection is of obvious importance. Their study was able to show the positive impact of socioeconomic development (marked decline in incidence before vaccine introduction to EPI, seen in figure 1), the further decline in incidence to a low level following HAV vaccine introduction, the increase in age of cases after vaccine introduction, and importantly, an immunity gap among children born just before HAV introduction into EPI (due to decreased force of infection following vaccine introduction). The study methods were appropriate. Their conclusions were based on evidence from their study.

Additional comments

I have a few suggestions to improve the manuscript.

There are some details missing about the serological survey. The manuscript does not say when the serological survey was conducted and which organization conducted the serological survey. From the description in the Ethical Review section, the serological survey was conducted for a different study. It appears that they authors obtain residual blood from that serological survey and analyzed samples for anti-HAV antibody levels. However, this is speculation on my part. The manuscript should be more clear. It would help the readers if the authors provided a reference to the survey. I’m assuming that informed consent was obtained for the original blood sampling and data collection.

The authors indicate that target sample sizes were calculated for each of the five age groups in the serological survey. They should include the sample size calculation in the methods section.

It seems that the most important findings of the study are the increased average age of cases and the large immunity gap among 10-15-year-old children. The authors describe the seroprevalence pattern as “unique” on line 186, but this pattern is not really unique, as it is also the national pattern according to their reference 4. Perhaps they could say that the pattern is characteristic of what is seen when highly effective vaccines are introduced without a wide-age-range catch-up campaign (rubella and hepatitis A are good examples). The authors give three plausible reasons for the immunity gap, and their third reason is probably the main reason – decreased infection rates in children reducing exposure to HAV in others.

The authors state on lines 234 and 235 that “evaluations that identify feasible approaches to mitigate the risk of HAV outbreak due to the immunity gap are urgent priorities.” This is a very reasonable and important recommendation. The authors could expand upon this recommendation by indicating actions they recommend taking or studies that need to be conducted prior to conducting a catch-up campaign, such as a health and economic impact modeling study of a campaign.

·

Basic reporting

The article is well structured. It is to be noted, however, that thorough linguistic revision is advised. Linguistic corrections include improper tense, punctuations, mixing between plural and singular and too long sentences. Also it is preferred to write it is (not it's) and did not (not didn't). Some lines needing revision are lines 28, 50, 75, 82, 110-112, 115, 136, 149, 153, 159, 175, 178-179, 192, 196, 230.

Experimental design

The experimental design is clearly presented.

Validity of the findings

In line 204, the authors refer to the results of a previous study the have conducted without mentioning them. It cannot be assumed that the audience have read that study.
In figure 1, kindly clarify what the blue area and the yellow area represent.
In table 1, the statistical significance of the seroprevalence among the age groups is not stated.

Reviewer 3 ·

Basic reporting

No comment

Experimental design

No comment

Validity of the findings

No comment

Additional comments

1. I strongly recommend to authors that the manuscript should be checked grammatically and re-read carefully.

2. There is a mess with the acronyms - some of them are introduced several times for examples.
For example: National Notifiable Disease Reporting System (NNDRS)

3. Table 2:
- Header row/column 2 should be mentioned in the table as No. (%).
- Ensure that the "Primary Symptoms (%)" and "Number of Asymptomatic Cases" rows in the table match the other sections by including the corresponding numbers and percentages.
- There is an error in the row "Initial Symptoms (%)." The percentage that indicated fever needs to be corrected.

4. The time period should be mentioned in the title of Table 1.

---

## Round 0.2 · accepted · Accept

All the reviewers now accepted the revised manuscript and I agreed with their recommendation.

Reviewer 1 ·

Basic reporting

I was a reviewer of the original manuscript. The authors have done a very good job addressing the few suggestions that I had to improve the manuscript. The manuscript is well structured, well written, and well referenced.

Experimental design

There are three studies reported - evaluation of surveillance data, a serological survey, and evaluation of recent reported cases of hepatitis A. The designs were appropriate and portray a comprehensive picture of the epidemiology over time and linked to policy changes.

Validity of the findings

This study is very important because it illustrates the impact and resulting epidemiological status of HepA vaccine introduction into a jurisdiction with a high prevalence of HAV infection. In such an epidemiological situation, WHO is cautious about HepA introduction, and this study provides an example of the results and impact of HepA introduction that will be useful to hepatitis experts and immunization programs in other countries.

Additional comments

The study is responsive to the 2022 WHO position paper on hepatitis A vaccines recommendations for research following HepA introduction. According to WHO, “Following the introduction of hepatitis A vaccines, it is important to regularly assess their impact using morbidity and mortality surveillance data. Seroprotection, and the duration of protection induced by single- and 2-dose schedules should be regularly monitored.”

Lines 85-87 – In the sentence, “This provincial serological study was conducted in 2017 by Shaanxi Provincial Center for Disease Control and Prevention and this provincial was designed to assess the prevalence of Hepatitis B,” there is a minor mistake, as the second “provincial” should instead be “study.”

·

Basic reporting

No comment

Experimental design

No comment

Validity of the findings

No comment

Additional comments

The authors have addressed the suggested corrections. However, in line 142 the authors still refer to their previous study without elaboration. I suggest they write:Serum samples and epidemiological data from reported cases were collected according to Hu et al (2021).

Reviewer 3 ·

Basic reporting

No comment

Experimental design

No comment

Validity of the findings

No comment